# Endogenous Mammalian Cardiotonic Steroids—A New Cardiovascular Risk Factor?—A Mini-Review

**DOI:** 10.3390/life11080727

**Published:** 2021-07-22

**Authors:** Natalia Słabiak-Błaż, Grzegorz Piecha

**Affiliations:** Department of Nephrology, Transplantation and Internal Medicine, Medical University of Silesia, Francuska 20/24, 40-027 Katowice, Poland; nataliablaz@gazeta.pl

**Keywords:** marinobufagenin, ouabain, hypertension, Na/K-ATPase, heart failure

## Abstract

The role of endogenous mammalian cardiotonic steroids (CTS) in the physiology and pathophysiology of the cardiovascular system and the kidneys has interested researchers for more than 20 years. Cardiotonic steroids extracted from toads or plants, such as digitalis, have been used to treat heart disease since ancient times. CTS, also called endogenous digitalis-like factors, take part in the regulation of blood pressure and sodium homeostasis through their effects on the transport enzyme called sodium–potassium adenosine triphosphatase (Na/K-ATPase) in renal and cardiovascular tissue. In recent years, there has been increasing evidence showing deleterious effects of CTS on the structure and function of the heart, vasculature and kidneys. Understanding the role of CTS may be useful in the development of potential new therapeutic strategies.

## 1. Basic Information, Structure, Biosynthesis of Cardiotonic Steroids, Mechanisms of Action

Cardiotonic steroids are a group of steroid hormones that circulate in the blood and are excreted in the urine. They are divided into two groups by their chemical structure: cardenolides (plant-derived) and bufadienolides (mainly of animal origin). Cardenolides have a five-membered unsaturated lactone ring attached to the steroid nucleus at position 17 and bufadienolides have a doubly unsaturated six-membered lactone ring (Figure 1). Cardenolides are isolated from plants, e.g., digitoxin from Digitalis purpurea, digoxin from Digitalis lanata and ouabain (g-strophanthin) from Strophanthus gratus and Acokanthera schimperi. Bufadienolides, such as marinobufagenin, telocinobufagin and bufalin, are isolated from the skin of Bufo toads; others, such as proscillaridin, can be obtained from plants of the genus Scilla and Drimia maritima [1]. Extracts from toad skin and plants, such as Digitalis, have been used for over 200 years in traditional/folk medicines for the treatment of congestive heart failure. To this day, some of the cardiotonic steroids, such as digitoxin, digoxin and proscillaridin, are still prescribed as a drug for heart failure. Recently, it has been discovered that endogenous cardiotonic steroids are also found in humans. In 1991, endogenous ouabain was isolated for the first time from human serum [2]. Ouabain was found in the serum of healthy volunteers but also in patients with congestive heart failure, hypertension and kidney failure [3]. Additionally, other CTS, such as marinobufagenin, telocinobufagin, the bufalin-like factor, digitalis-like factor, and proscillaridin A, were found in the human body. For example, marinobufagenin was discovered in the urine of patients with myocardial infarction and in the serum of patients with advanced chronic kidney disease [3]. The synthesis of cardiotonic steroids from cholesterol takes place in the glomerular and fascicular layer of the adrenal glands [4,5], but it was discovered that CTS may be released also from the placenta [6], heart [7], and hypothalamus [8]. The biosynthesis of cardiotonic steroids is unclear and it seems to be complex. Thus far, it was found that marinobufagenin and ouabain are released in response to the adrenocorticotropic hormone, angiotensin II via the AT2 receptor, vasopressin, phenylephrine, volume expansion, increase in physical activity, hypoxia, and behavioral stress [9,10,11]. In animal experiments, it was shown that with an infusion of 0.9% saline or feeding with a high-sodium diet also results with an increase in CTS levels in blood [12,13,14]. 

The biosynthesis of endogenous CTS in humans remains poorly understood. Biosynthesis of ouabain and dihydro-ouabain has been demonstrated in human cultured adrenocortical cells [15].

Until recently, it was believed that CTS act mainly as inhibitors for the ubiquitous transport enzyme called sodium–potassium adenosine triphosphatase (Na/K-ATPase) [1]. Na/K-ATPase maintains the gradient for both the sodium and potassium concentration across the cell membrane in almost every cell. Therefore, Na/K-ATPase plays a key role in many basic, physiological processes, for example, cell volume regulation, and the transport of ions, glucose and amino acids. In the neuronal cell, Na/K-ATPase maintains and restores the membrane potential required for excitability. In renal tubular cells, it is necessary for the regulation of sodium reabsorption from glomerular filtrate, which is the key in controlling extracellular volume and blood pressure [1]. The activity of Na/K-ATPase drives also the Na/Ca exchanger and, thus, regulates the concentration of calcium in cytosol, the latter being the second messenger for many cell functions. In muscle cells, including cardiomyocytes and vessel smooth muscle cells, an elevated cytosol level of calcium leads to an increase in contractibility which may also result in a positive inotropic effect, increasing tension of the vessel wall and increasing blood pressure [1]. This classical “ionic pathway”, through which CTS inhibit Na/K-ATPase activity in renal tubules, promotes natriuresis. It is a physiological function in response to sodium-induced volume loading and may act as a mechanism for lowering blood pressure. On the other hand, the inhibition of Na/K-ATPase by CTS in blood vessel smooth muscle cells leads to vasoconstriction. Therefore, a possible, physiological role of endogenous CTS is the regulation of sodium content, blood volume, and pressure. The classical pathway of CTS is also responsible for the positive inotropic effect of CTS, which has also been observed in patients treated with digoxin. The inhibition of Na/K-ATPase by CTS in cardiomyocytes leads to an increase in the sodium concentration and a secondary increase in the calcium concentration by influencing the Na/Ca exchanger, resulting in an increase in the force of contraction of the cardiac muscle. Additionally, it was observed that CTS decrease Na/K-ATPase activity not only by the direct inhibition of the Na-K pump, but also by decreasing the presence of plasmalemmal Na/K-ATPase. Studies, both in vivo and in vitro, showed that an increasing level of MBG after a sodium load, induces endocytosis and the internalization of plasmalemmal Na/K-ATPase in renal tubule cells, decreasing both the basolateral and apical sodium transport. The administration of an antibody to MBG, blocked both the endocytosis of Na/K-ATPase and natriuresis [16].

It is now believed that Na/K-ATPase, apart from its role as a Na^+^ and K^+^ pump, has also an additional, signaling function. Na/K-ATPase acts as a receptor for CTS and takes part in the cell signal transduction pathways [17]. It has been shown that some portion of Na/K-ATPase is located in the caveolae and it is physically associated with other signaling proteins (Src kinase, epithelial growth factor receptor (EGFR), phospholipase C, Phosphoinositide 3-kinase, caveloin-1). On the extracellular loop of the Na/K-ATPase alpha subunit, there is a specific binding site for CTS. Binding CTS to this receptor complex activates several downstream signaling cascades, which lead to the activation of genes responsible for cell growth, proliferation, differentiation, adhesion and may others metabolic pathways. For example, ouabain binding to the Na/K-ATPase complex in cardiomyocytes activates the EGFR/Src/ERK and phosphoinositide 3 kinase pathway which results in the hypertrophic growth of cardiomyocytes [18]. It is currently postulated that the effect of CTS on the cellular signaling function of Na/K-ATPase plays an important role in the development of kidney or heart failure. 

Na/K-ATPase is composed of three subunits: alpha, beta and FXYD [1]. The alpha subunit, comprising ten transmembrane helices, is the catalytic subunit and contains binding sites for ATP, CTS and other ligands. The single transmembrane beta subunit is essential for enzymatic activity, modulates the enzyme affinity to Na+ and K+ ions and it also functions as a chaperone targeting the α subunit to the plasma membrane. FXYD is an auxiliary subunit that regulates Na/K-ATPase activity in a tissue- and isoform-specific way. There are four alpha, three beta and seven FXYD isoforms. This allows the creation of numerous different combinations of alpha–beta complexes, in particular in tissues with different sensitivity to CTS. Cardenolides (for example ouabain) act mainly on the alpha 2 isoform, which is widely expressed in the heart, smooth cells, skeletal muscle, brain, adipocyte, cartilage and bone, and on the alpha 3 isoform, which is widely expressed in excitable tissue. Their effect on the alpha 1 isoform is much weaker in rodents, [19] but seems to be much more pronounced in humans [20]. Bufadienolides (for example marinobufagenin) have a predilection to the ubiquitous alpha 1 isoform, which is also a dominant isoform in the kidney. Marinobufagenin (MBG) promotes natriuresis through the inhibition of Na/K-ATPase in renal proximal tubules, but ouabain does not have natriuretic properties. MBG, in a concentration insufficient to inhibit the pumping mechanism of Na/K-ATPase, promotes natriuresis also by initiating the signaling pathway and promotes the internalization of Na/K-ATPase in the proximal tubule and decreased expression of the transport protein, Na/H exchanger, in this tubule. MBG also promotes vasoconstriction by inhibiting Na-K/ATPase in vascular smooth muscle cells. Ouabain does not promote natriuresis but probably has a role in the adaptation to sodium depletion and loading. An increased salt intake stimulates the synthesis of MBG, which is preceded by a transient, short-term ouabain increase. It was also observed that after 2 weeks of sodium depletion, the serum concentration of ouabain increases. It is postulated that ouabain may act as a neurohormone, triggering the release of MBG, which in turn increases the contractility of the heart, vasoconstriction and natriuresis [17].

## 2. CTS in Pathophysiology of the Cardiovascular System

### 2.1. Hypertension

Apart from the physiological role of CTS in the regulation of sodium content, blood volume and pressure, CTS also take part in the pathological adaptation resulting in hypertension. Experiments in animals show that the infusion of ouabain or marinobufagenin, at concentrations similar to endogenous levels, leads to an increase in blood pressure [21,22]. Increased plasma levels of CTS (MBG or ouabain) are particularly prominent in states of volume expansion and were observed in humans with essential hypertension, primary hyperaldosteronism [23], renal artery stenosis [24], chronic kidney disease [25,26], preeclampsia [27], or heart failure [28,29]. The relevant association between MBG and elevated blood pressure has been proven by the administration of the anti-MBG antibody to the Dahl salt-sensitive hypertensive rat (a rodent model of salt-sensitive hypertension), which caused a decrease in systolic blood pressure [30].

The relationship between salt, and especially salt-sensitivity, and blood pressure is well known, and has been demonstrated by several large clinical trials, such as the International Study of Salt and Blood Pressure (INTERSALT) [31] and the Dietary Approaches to Stop Hypertension (DASH) study [32]. A high dietary salt intake is associated with an increased blood pressure and vascular stiffening due to an altered endothelial and vascular smooth muscle cells (VSMCs) function and extended arterial wall fibrosis [33,34,35]. A high sodium diet results in a faster pulse wave velocity (PWV), indicating arterial stiffening, which precedes the development of hypertension with aging [36,37,38].

Observations from studies, both in animals and humans, have shown that an increased salt intake associates with elevated MBG levels. Studies, both in rats by Fedorova et al. [39] and in humans by Strauss et al. [40], suggest that increased salt intake promotes an increase in the pituitary endogenous ouabain, stimulating the renin–angiotensin–aldosterone system and sympathetic activity, which in turn stimulates the synthesis of MBG in adrenal glands. Since studies by Dahl [41,42] and also by de Wardener and Clarkson [43], there has been a hypothesis that a circulating “humoral factor” induces salt-sensitive hypertension. Now, it has been proposed that CTS are these “humoral factors” and play an important role in the pathogenesis of salt-sensitive hypertension. CTS were shown to exhibit both natriuretic as well as vasoconstrictive properties via the inhibition of renal and vascular Na+/K+-ATPase. Interesting data have come from a study by Bagrov et al. [44] with two different strains of rats: normotensive Sprague Dawley (SD) rats and Dahl salt-sensitive (DS) rats. Sodium loading in normotensive (SD) and salt-sensitive (DS) rats generated an increase in plasma levels and renal MBG excretion, as well as different patterns of sodium pump inhibition in the vascular smooth cell and the renal medulla. In DS rats, the aortic sodium pump was more inhibited compared to normotensive SD rats, which in turn showed a greater inhibition of the renal sodium pump. An increase in blood pressure was observed in DS rats, while no pressor response in the SD rats. On the other hand, the normotensive SD rats showed a greater natriuretic response than DS rats. Inefficient natriuresis in salt-sensitive rats may be associated with a reduced Na/K-ATPase sensitivity to MBG in kidneys compared to normotensive Sprague Dawley rats [44,45]. In vitro studies have revealed that MBG acts as a vasoconstrictor to pulmonary artery rings in a concentration-dependent manner [46]. The administration of an anti-MBG antibody to Dahl salt-sensitive rats and in NaCl-supplemented pregnant rats with elevated MBG levels was associated with the restoration of vascular sodium pump activity and reduction in blood pressure. This observation indicates that MBG exerts its pressor effects by inhibiting Na/K-ATPase in vascular smooth muscle [47]. Taken together, these data suggest that the alteration in natriuresis and increase in vasoconstriction may contribute to the etiology of salt-sensitive hypertension. Additionally, kidney dysfunction not only diminishes the natriuretic function of MBG, but also causes an additional stimulation of MBG production and increases vasoconstrictive properties of MBG. This process may initiate and accelerate a vicious cycle of salt sensitivity.

CTS can also act on Na/K-ATPase as a hormone that stimulates its signal transduction. It was observed in rats, in vivo and in primary cultured human and rat artery myocytes, that a sustained exposure of arterial myocytes to ouabain activates an α2 Na/K-ATPase-mediated protein kinase (PK) signaling cascade. This leads to an increased expression of Ca^2+^ transporters, including NCX1 and SERCA, which promote a long-term arterial Ca^2+^ gain and sequestration in the sarcolemmal reticulum and lead to the long-term elevation of blood pressure [48]. In the study by Fedorova et al., the incubation of aortic rings and vascular smooth muscle cells in the presence of MBG resulted in a reduction in the Fli-1 levels, a concomitant increase in the collagen-1 levels, and an increase in vascular wall fibrosis [49]. It has also been observed that in MBG pre-treated aortic rings that were precontracted with endothelin-1, the response to the relaxant action of sodium nitroprusside was significantly reduced compared to vehicle pre-treated rings. The addition of canrenone, a metabolite of spironolactone to incubation, prevented MBG-induced collagen synthesis, and the sensitivity of vascular rings to sodium nitroprusside was restored. Canrenone alone was unable to induce any previous changes in the rings of the vascular aorta [49]. Grigorova et al. found that in rats fed for 8 weeks on a high-sodium diet, the increased level of MBG was associated with an increased velocity of the pulse wave and collagen in the aortic wall, and a greater expression of aortic TGFβ1, collagen 1α2 and collagen 4α1 [50]. All these parameters were reversed when the normal salt diet was restored. MBG stimulation of cultured vascular smooth muscle cells (VSMC) has also been shown to increase the production of collagen-1 and TGFβ-1 by these cells and is associated with the activation and phosphorylation of the Src and PKCδ proteins, which are associated with the activation of several signaling pathways, including signaling the profibrotic pathway dependent on Fli1 [50].

Strauss et al. demonstrated that in young adults, MBG excretion is also associated with an increased blood pressure [51] and arterial stiffness [52]. It has also been shown that elevated MBG levels in young, apparently healthy adults with normal blood pressure, are associated with an increase in pulse wave velocity independent of sodium intake [52]. These results suggest that MBG may contribute to sodium- and pressure-independent vascular changes in people without known cardiovascular disease. It has recently been postulated that arterial stiffness may not only be a complication of hypertension, but may also play a precursor role in the development of hypertension.

Considering the above, MBG may become an early biomarker of increased cardiovascular risk.

### 2.2. Cardiovascular Diseases

Increased levels of cardiotonic steroids, such as MBG or ouabain, have been documented in a variety of cardiovascular disease states, among others, in congestive heart failure [28,29] and myocardial infarction [53]. In animal-sustained ouabain [54] or MBG [21] infusion in rats, induced left ventricular hypertrophy. It is now believed that cardiotonic steroids may also have a deleterious effect in the cardiovascular system, which is independent of its effect on blood pressure. Fedorova et al. demonstrated that in normotensive rats, a high-salt intake stimulates MBG production and tissue remodeling in the heart and kidneys, even in the absence of a blood pressure increase [12]. Experiments by Grigorova et al. showed that elevated MBG levels in young normotensive rats in response to a high-salt intake increased vascular fibrosis and induced impaired vasorelaxation in the absence of hemodynamic changes. It was also shown that the administration of an anti-MBG antibody and the neutralization of MBG reduced vascular fibrosis and improved vascular relaxation, which was also independent of blood pressure changes [55]. In another set of experiments, Grigorova et al. showed that a high-sodium diet in young normotensive rats is associated with an increase in MBG accompanied by an activation of TGF-β signaling, aortic fibrosis and aortic stiffness. A reduction in dietary salt in these rats decreased MBG levels, pulse wave velocity, aortic wall collagen content and expression of TGF-β. This observations suggest that a decrease in salt intake reduces MBG, which results in restoring aortic elasticity and may diminish the risk of cardiovascular disease [50]. Experiments, both in vivo and in vitro, demonstrated that the administration of MBG directly stimulates cardiac fibroblasts to produce increased amounts of collagen through the Na/K-ATPase signal cascade [56]. Dahl salt-sensitive (Dahl-S) rats on a high-sodium diet developed hypertension, compensatory left ventricular hypertrophy, congestive heart failure, increased synthesis, excretion of MBG and activated cardiovascular and renal TGFβ-dependent profibrotic signaling. Zhang et al. showed that the administration of an anti-marinobufagenin monoclonal antibody decreased the expression of profibrotic genes, decreased aortic stiffness, reduced heart mass, restored cardiac function estimated by echocardiography, decreased renal fibrosis, kidney weight, reduced collagen I, III, and IV, and downregulated TGFβ profibrotic signaling in the kidneys. It was also shown in vitro that a 24-hour treatment of cardiomyocytes with physiological, nanomolar concentrations of marinobufagenin stimulated TGFβ and Fli-1-depending pro-fibrotic signaling in cardiomyocytes [57]. CTS may also take part in the pathophysiology of heart remodeling by influencing cell death and growth in heart tissue. It was observed that patients with heart failure had a decreased cardiac Na/K-ATPase expression on a heart biopsy. The reduction in Na/K-ATPase α1 correlates with the cardiac contractility deficiency in animal models and in humans. The reduction in cellular Na/K-ATPase causes cell growth inhibition and cell death. Liu et al. examined if a reduction in Na/K-ATPase in combination with increased cardiotonic steroids may cause cardiac myocyte death and cardiac dysfunction [58]. They found that MBG infusion increased myocyte apoptosis and induced significant left ventricle dilation in Na/K-ATPase α1 heterozygote knock-out mice (α1+/−), but not in their wild-type littermates. The infusion of MBG in wild-type mice activated the Src/Akt/mTOR signaling pathway, which further increased the phosphorylation of ribosome S6 kinase (S6K) and BAD (Bcl-2-associated death promoter), which protected myocytes from apoptosis. In contrast, in α1+/− mice myocytes, MBG failed to induce this signaling pathway. Moreover, it also caused the activation of caspase 9 and, finally, induced myocyte apoptosis [57]. These results indicate that a reduction in Na/K-ATPase enables MBG to induce cardiac myocyte apoptosis, leading to heart failure. Taking into account that in heart failure both an increased level of MBG as well as a reduction in Na/K-ATPase expression in cardiomyocytes were observed, this mechanism may be one of the pathways responsible for the development and progression of heart failure.

Chronic kidney disease (CKD) increases cardiovascular (CV) risk immensely. CV disease is the leading cause of death in CKD and end-stage renal disease patients. However, the nature and pathophysiology of CV disease in CKD is complex and both classical (e.g., arterial hypertension, dyslipidemia) and non-classical (e.g., inflammation, vascular calcification) factors take part. It was shown that both rats receiving MBG (to achieve concentration similar to those observed after partial nephrectomy) as well as rats after partial nephrectomy (which is an animal model for chronic kidney disease) developed a very similar increase in cardiac weight and fibrosis, increased systemic oxidative stress and a decrease in left ventricular vasorelaxation. It was also discovered that these changes were attenuated if rats were immunized against MBG. It is worth to say that only rats after partial nephrectomy had an increased aldosterone concentration and the administration of MBG in rats without nephrectomy did not influence serum aldosterone concentration. These data show that MBG’s effects on the heart are not mediated via increasing the aldosterone synthesis [21]. Spironolactone and its major metabolite—canrenone—, apart from its ability to antagonize aldosterone binding to the mineralocorticoid receptor, also interacts with the plasmalemmal sodium–potassium ATPase (Na/K-ATPase). Tian et al. observed how spironolactone and canrenone therapy in rats may influence cardiomyopathy induced by either experimental renal failure or by the administration of MBG (mimicking levels of MBG observed after partial nephrectomy) [59]. It was found that spironolactone and canrenone attenuated the diastolic dysfunction as assessed by cardiac catheterization and essentially eliminated cardiac fibrosis as assessed by histological analysis and Western blot. The administration of spironolactone did not alter either the plasma concentrations of MBG or aldosterone, but it prevented MBG signaling through Na/K-ATPase, inhibiting the ERK-1/2 and protein kinase C signal transduction [59]. These data suggest that spironolactone appears to competitively inhibit MBG signaling and the stimulation of collagen production in vitro, which may partly explain the beneficial effects of spironolactone in vivo.

Interesting data were shown in the analysis by Strauss et al. in the African-PREDICT study, performed in a young population (aged 20–30 years), free of detected cardiovascular or other chronic diseases. They found a positive, blood pressure-independent association of the left ventricular mass normalized for body surface area, the most sensitive estimate of subclinical cardiac target organ damage, with 24-hour urinary MBG excretion in young adults with excessively high levels of MBG excretion. These findings suggest that elevated levels of MBG, particularly in women, may increase the risk of the development of future cardiovascular disease [60]. It is also postulated that the adverse effects of MBG on cardiac remodeling may be exacerbated by obesity, due to an increased sensitivity of Na+/K+-ATPase to MBG. Strauss-Kruger et al. observed 275 healthy young participants over 4.5 years, and discovered that baseline MBG excretion levels were associated with an increased left ventricular mass index over time in obese individuals [61]. Pitzalis et al. demonstrated that a higher serum concentration of ouabain predicted a more severe progression of heart failure [62].

## 3. Conclusions

Salt sensitivity is, as a cardiovascular risk factor, independent of blood pressure [63] and it is associated with an increased mortality not only in hypertensive, but also in normotensive adults [64]. CTS are postulated to be an important factor connecting salt intake, salt sensitivity and cardiovascular diseases and they may become biomarkers or indicators of future cardiovascular complications. At this time, experimental data show that CTS may contribute to end-organ damage independent of blood pressure (Figure 2). If this could be reproduced in humans, CTS may represent a new target for therapy.

## Figures and Tables

**Figure 1 life-11-00727-f001:**
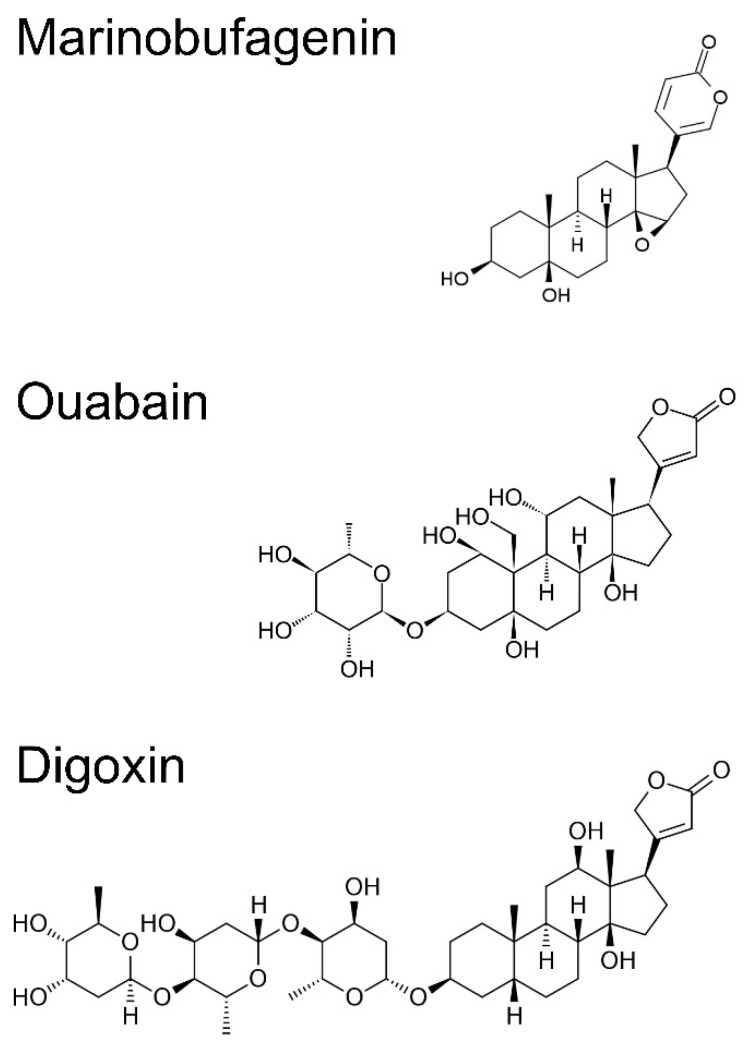
Chemical structure of marinobufagenin, ouabain, and digoxin.

**Figure 2 life-11-00727-f002:**
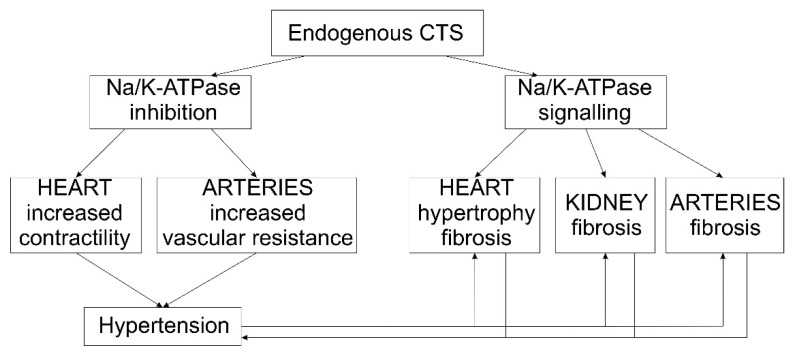
Proposed mechanisms linking CTS, hypertension and end-organ damage.

## Data Availability

Not applicable.

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
