# Peer review of "Endogenous Mammalian Cardiotonic Steroids—A New Cardiovascular Risk Factor?—A Mini-Review"

_life, 2021, doi:10.3390/life11080727_

Round 1
Reviewer 1 Report
This manuscript reviews the impact of endogenous mammalian cardiotonic steroids (CTS) in the cardiovascular system and discussed about the potential pathophysiological role they are playing. This review comes out in a very timely manner and I believe it should attract many readers of the journal. There are several issues to be fixed before it can be accepted.
- The first paragraph talked about the chemical structures of the two groups of CTS, i.e. cardenolides and bufadienolides. I recommend adding a figure showing the exact structures of these compounds, especially ouabain, marinobufagenin, and digoxin as these three are often mentioned. Put the chemical structures side by side will be much easier for the future readers to appreciate the similarity as well as the difference in their structures.
- In line 103, the sentence “Cardenolides (for example ouabain) act mainly on alfa 2 isoform” is not accurate. Many papers have shown that ouabain binds to alpha1 and stimulates signaling function of alpha1. So this sentence needs to be modified.
- There are many typo errors throughout the manuscript. For example, in many sentences the authors use “alfa” instead of “alpha”. To be consistent, I suggest the authors carefully double check the text and change all “alpha” or “alfa” to the Greek letter “a”. In line 123, “CTS take also part…” should be “CTS also take part…..”.
- I believe the notion of “endogenous CTS” is still under debate because it is still not clear how those CTS compounds were synthesized? It would be helpful to list the open questions to be answered before endogenous CTS can be confirmed as cardiovascular risk factors?
Author Response
We thank the reviewer for these helpful suggestions.
- We added a figure presenting the chemical structures of various CTS.
- We amended the information on ouabain ant its effect on alpha 1 Na/K-ATPase.
- We re-checked and corrected the typing errors.
- We stressed out that the synthetic pathway by which CTS are produced in mammals are not fully understood, despite detection of these compounds in various body fluids and tissues.
Reviewer 2 Report
The review is timely and overall well‐reasoned, and well written. Following are my comments which authors can address to further improve the current manuscript.
- This review can be further complemented by including Graphical representation illustrating the proposed biosynthesis or function of endogenous CTS
- Authors might consider discussing the following reference in review section 2.2
Pitzalis MV, Hamlyn JM, Messaggio E, Iacoviello M, Forleo C, Romito R, de Tommasi E, Rizzon P, Bianchi G, Manunta P: Independent and incremental prognostic value of endogenous ouabain in idiopathic dilated cardiomyopathy. Eur J Heart Failure 2006;8:179-186.
Minor
The manuscript needs a careful revision to address the typo.
Author Response
We thank the reviewer for these helpful suggestions.
We included a figure presenting proposed mechanisms linking CTS, hypertension and end-organ damage.
We also discussed the suggested reference.
The manuscript has been checked for errors.